# Overcoming Barriers to Wound Healing in a Neuropathic and Neuro-Ischaemic Diabetic Foot Cohort Using a Novel Bilayer Biodegradable Synthetic Matrix

**DOI:** 10.3390/biomedicines11030721

**Published:** 2023-02-27

**Authors:** Frank P. Guerriero, Robyn A. Clark, Michelle Miller, Christopher L. Delaney

**Affiliations:** 1Department of Vascular and Endovascular Surgery, Flinders Medical Centre, Bedford Park, Adelaide, SA 5042, Australia; 2College of Nursing and Health Sciences, Flinders University, Bedford Park, Adelaide, SA 5042, Australia; 3College of Medicine and Public Health, Flinders University, Bedford Park, Adelaide, SA 5042, Australia

**Keywords:** diabetic foot ulcer, wound healing, biodegradable synthetic matrix

## Abstract

Diabetes-related foot ulceration presents an increasing risk of lower limb amputation globally, driving the search for new treatment technologies. Our single-centre prospective observational study reports on the impact of bilayer biodegradable synthetic matrix technology (NovoSorb^®^ BTM) on the healing and amputation rates of a diabetic foot ulceration cohort. Consecutive patients with a diabetes-related foot ulceration treated with NovoSorb BTM, between December 2019 and October 2021, were followed for 12 months. Complete wound healing and amputation outcomes were observed. Amputation risk was stratified using the Wound, Ischaemia and foot Infection (WIfI) classification system. Study outcomes were compared with recently published meta-analysis data to evaluate the impact of the synthetic matrix. In total, 25 NovoSorb BTM applications to 23 wounds in 22 patients were observed. Complete wound healing was observed in 15 of the wounds, 3 retained chronic wounds, 3 required minor amputation and 2 required major limb amputation. Further, 12-month WIfI amputation risk analysis saw 18 patients stratified to WIfI stage 4, 4 to WIfI stage 3 and 1 to WIfI stage 1. Our observed 12-month major amputation rates were 11.1% (*n* = 2) for stage 4 and 0% for stages 3 and 1. Our early experience suggests that NovoSorb BTM is a safe and effective treatment for moderate to severe diabetes-related foot ulceration. While larger-scale data are required, NovoSorb BTM may represent a promising new addition to the armamentarium of clinicians, who strive to achieve limb salvage in this complex cohort of patients.

## 1. Introduction

Lower limb amputation is a complication of foot ulceration in patients with diabetes and/or peripheral arterial disease. The annual incidence of foot ulceration in patients with diabetes is reported to be between 2% and 5%, with a lifetime risk of 15% to 20%. With the global prevalence of diabetes is reported to be 415 million [1], with conservative estimates placing the global incidence of diabetes-related foot ulceration at around 8.3 million. Furthermore, observational studies place rates of amputation in patients with diabetes-related foot ulceration between 6% and 43% [2]. Up to 3.5 million amputations may, therefore, be required globally on an annual basis. In fact, throughout the world, it is estimated that every 30 s, a leg is amputated, and 85% of these amputations are the result of a foot ulcer [3].

It is, therefore, understandable that the concept of amputation prevention has become a growing focus of vascular surgical practice. While advances in endovascular revascularisation techniques have provided improvements in treating macrovascular ischaemia, the associated micro-ischaemia and compromised immuno-cellular environment remain, which stand as barriers to wound healing [4]. Despite radiologically optimal outcomes, many patients do not reach the perfusion pressure threshold for adequate wound healing.

The combination of ischaemia, neuropathy, and immuno-compromise in diabetes-diseased tissue presents a significant challenge to wound healing, often limiting the success of traditional surgical approaches for assisted wound closure, such as split thickness skin grafting and autologous flaps. Use of emerging augmentative wound closure technologies, such as exografts, allografts and acellular dermal matrices, in the treatment of diabetes-related foot ulceration continues to be evaluated and evidence concerning long-term outcome data is lacking [5]. The current limitations of assisted wound closure options for vascular disease patients expose this cohort to extended healing time frames, higher risk of infection and, subsequently, a higher risk of limb amputation.

The proliferative properties of a novel bilayer biodegradable synthetic matrix product (NovoSorb^®^ BTM, PolyNovo Biomaterials Pty Ltd., Port Melbourne, VIC, Australia) are suitable for application over exposed deeper structures, such as tendon and bone [6,7,8], and may be a new alternative for assisted wound closure in patients with diabetes-related neuropathic and neuro-ischaemic foot wounds. BTM is a 2 mm-thick biodegradable polyurethane foam matrix, which houses a pre-fenestrated non-biodegradable sealing membrane on the outward-facing surface (Figure 1a). The sealing membrane serves to physiologically close the wound and protect the underlying matrix during the proliferation of tissue into the open structure foam matrix (integration) during formation of a vascularised neo-dermis (Figure 2). As cellular migration begins into the foam matrix, the chambers are infiltrated by a variety of cell types with the interconnecting pores, allowing for the exchange of nutrients and waste (Figure 1b). Most critically, the chambers compartmentalize the wound response, minimizing the foreign body response and creating a microwound that the body can heal through regeneration. Once tissue proliferation throughout the foam matrix is complete, the sealing membrane is removed to allow epithelialisation to occur [7]. NovoSorb® BTM (BTM) is fully synthetic, contains no biological molecules and retains its structure while it is slowly broken down by hydrolysis within the integrating tissue [7].

This single-centre prospective observational study reports on the impact of bilayer biodegradable synthetic matrix technology (NovoSorb^®^ BTM) on the healing and amputation rates of a complex diabetic foot ulceration cohort.

## 2. Materials and Methods

### 2.1. Study Design

This single-centre prospective observational study investigated the impact of BTM on the proliferation of tissue for the purpose of wound healing and limb salvage in a neuropathic and neuro-ischaemic diabetic foot wound cohort.

#### Ethics

The study was approved by the by the Ethics Committee of Southern Adelaide Local Health Network Office for Research (approval number HRE0214/2021). 

### 2.2. Participants

Consecutive patients treated between December 2019 and October 2021 were prospectively observed if they met the following criteria:A confirmed diagnosis of diabetes, as determined by blood glycosylated haemoglobin (HbA1C) >6.5% (48 mmol/mol) or as represented by pre-existing prescription of diabetes medication (e.g., insulin or oral hypoglycaemics) within 12 weeks of admission;Who were 18 years or older;Hosted a wound in the foot, distal to malleoli, arising from either surgical amputation or debridement of a chronic wound;Who were treated with BTM for the purpose of tissue reconstruction, treatment of tissue deficit or exposed deeper structures, such as bone and tendon, were included.

### 2.3. Study Outcome

Complete wound healing, defined as 100% epithelialisation with no exudate from the site of original BTM application, was established as the primary outcome. Secondary outcomes were time (days) to complete wound healing, occurrence of wound infection post-BTM application and major and minor amputation post-BTM application. Outcome data were collected prospectively and evaluated at 12 months post-date of surgical application of BTM.

### 2.4. Technique

Patients undergoing BTM application were optimised with maximal revascularisation prior to application. The supervising clinician would determine that the wound was clinically free from infection prior to BTM application. BTM was applied in an operating theatre environment. The wound bed was surgically debrided to healthy bleeding tissue, removing any non-viable tissue from the wound bed prior to application. Although pre-fenestrated, additional fenestrations (5–10) were made to the BTM prior to application using the tip of a scalpel on an inverted kidney dish, to ensure adequate extraction of wound exudate through the sealing membrane. The BTM was cut to match the size of the wound bed using scissors and pressed against the bleeding tissue to visibly moisten the foam with blood from the wound. The BTM was secured to the wound bed using either surgical staples or sutures to facilitate apposition of the product with the wound bed (Figure 1).

Intraoperatively, the BTM was further bolstered into the wound bed utilising polyurethane foam dressing (Granufoam™, 3M/KCI San Antonio, USA) and secured in place with clear polyurethane adhesive drape and Negative Pressure Wound Therapy (NPWT) applied (VAC™, 3M/KCI San Antonio, TX, USA) with a continuous pressure of −75 mmHg for an initial 5–7 days.

After initial post-surgery review (5–7 days), NPWT was reapplied for a further 7 days as per the above method, to further bolster the BTM into the wound bed and manage wound exudate. There was no contact of the Granufoam™ with the wound bed—it sits atop the BTM sealing membraned for all NPWT applications.

Post the initial 14 days of NPWT, management of the in situ BTM continued using topical antimicrobial wound dressings (povidone iodine and paraffin-impregnated gauze) bolstered with sterile gauze and a sterile composite absorbent dressing, secured with hypoallergenic fixation tape or crepe bandage. Wound dressings were replaced a minimum of 3 times a week, more frequently in the event of high volumes of wound exudate.

The sealing membrane and surrounding skin were sprayed with a topical hypochlorous acid (Microdacyn^®^ TeArai Biofarma, Herne Bay, Aukland, New Zealand) post-removal of previous dressing, at each dressing change, to reduce risk of infection.

Delamination of the BTM was performed once the sealing membrane layer could be easily separated from the underlying integrated tissue. Securing medium (staples or sutures) was left in situ until day of delamination.

Post-BTM delamination management was facilitated through conventional wound management practice, utilising principles of wound bed preparation [9], infection prophylaxis [10] and biomechanical offloading [11]. Assisted wound closure through split thickness skin grafts is not typically employed for diabetic foot ulcers in our surgical unit due to higher risk of post-operative complications, delayed healing times and additional exposure to anaesthetic [5,12].

Patients received a collaborative mixture of community, podiatric and vascular surgery outpatient follow-up, with a minimum of monthly attendance to a multidisciplinary vascular surgery outpatient service. 

### 2.5. Safety

Patients were monitored for adverse events related to application of BTM.

### 2.6. Data Collection

Demographics, including patient age and sex, were obtained from the medical record. Pre-procedural HbA1C was also collected. 

### 2.7. Primary Outcome Data

Wound healing outcomes were observed at 12 months post the nominated inclusion period (December 2019 through October 2021). 

Cases where complete wound healing was not observed were classified as chronic wounds (not healed). 

### 2.8. Secondary Outcome Data

Time (days) to complete wound healing was calculated using the difference between date of BTM application and documented date of complete epithelialisation with no exudate, taken from the follow-up visit date where complete epithelialisation was observed.

Those who underwent amputation post-application of BTM were stratified into minor and major amputations, defined as per the International Working Group of the Diabetic Foot (IWGDF) ‘definitions and criteria for diabetic foot disease’ document [13].

Data concerning wound infection pre- and post-BTM application were collected via medical record documentation for the admission where BTM application occurred. For this study, infection was classified using the Society for Vascular Surgery (SVS) adaptation of the Infectious Diseases Society of America (IDSA) and IWGDF Guidelines on the diagnosis and treatment of foot infection in persons with diabetes [10,14]. All confirmed wound infections were systemically treated with appropriate antibiotic regime and the wound determined clinically free from infection by the treating clinician, prior to BTM application.

Pre-BTM application toe pressures were obtained post-revascularisation procedures and prior to BTM application. A toe pressure >50 mmHg was established as a predictor of wound healing [15]; however, this did not serve as a cut-off for determining application of BTM but helped to inform ischaemia severity.

### 2.9. Data Analysis

Continuous data were reported as mean (SD) or median (IQR) according to normality and categorical data as *n* (%). Differences within groups for continuous data were tested using paired samples *t*-test. Differences between groups for categorical data were tested using χ^2^ test. A statistical significance level of *p* < 0.05 was utilised. IBM SPSS was utilised for statistical testing (IBM Corp. Released 2016. IBM SPSS Statistics for Windows, Version 24.0. IBM Corp, Armonk, NY, USA).

### 2.10. Predicted Amputation Risk in the Observed Cohort

To illustrate the severity of the wounds treated with BTM, the predicted 12-month risk of amputation, prior to BTM application, was determined using the validated Wound, Ischaemia and foot Infection (WIfI) classification system [14]. The score was calculated with wound severity and infection severity grading based on pre-procedural clinical photography, pre-procedural perfusion tests (after revascularisation) (i.e., toe pressures), pre-procedural signs of infection as per IDSA guidelines and clinical documentation. Patient lower limb 12-month amputation risks were stratified into Very Low (stage 1), Low (stage 2), Moderate (stage 3) or High (stage 4) categories, as per the WIfI classification system [14].

The assigned WIfI classification grading was employed to compare wound healing outcomes and 12-month amputation rates of BTM-treated patients with expected rates of amputation based on the widely validated global WIfI data in a recently published meta-analysis [16]. WIfI classification was also used to compare outcomes between wounds originating from either surgical amputation or pre-existing chronic wounds, treated with BTM.

## 3. Results

During the nominated study period (December 2019 to October 2021), we observed a total of 25 BTM applications to 23 wounds amongst 22 patients treated for diabetic foot wounds. Out of the 22 cases, 4 underwent more than one application of BTM—3 who underwent repeat application to their originally treated wound bed and 1 who had a new wound on the opposite foot that was also treated with BTM (Table 1).

The majority was male (*n* = 19, 86%), with a mean age of 67.3 years (SD 10.2) (Table 1).

### 3.1. Wound Healing and Amputation Outcomes

A total of 15 (65.3%) of 23 wounds progressed to complete wound healing post-BTM application (Table 2). Of the remaining eight non-healed wounds, three retained a chronic wound, three required minor amputation at the site of BTM application and two required major amputation of the BTM-treated limb, both secondary to wound deterioration and significant calcaneal bone loss related to refractory osteomyelitis (Table 2).

Of the three patients with a chronic wound, two experienced a greater than 60% reduction in wound surface area, observed during the follow-up period. The remaining patient passed away due to cancer-related illness prior to complete wound closure.

Of the three patients who underwent minor amputation of the BTM-treated area, two progressed to complete wound healing and one retained a chronic wound.

### 3.2. Time to Complete Wound Healing

Of the 15 wounds that achieved complete wound healing, average time to complete wound healing was 158 days (median 126, IQR 81).

### 3.3. Pre- and Post-BTM Application Wound Infection

Out of the 23 wounds, 12 (52.1%) were classified as hosting signs of infection, ranging from mild through to severe, within 14 days prior to BTM application. All 12 wounds received treatment for wound infection prior to BTM application. Of these, six progressed to complete wound healing, three progressed to minor amputation, two to major amputation and two patients retained a chronic wound at the end of the observation period.

A total of three (13%) wounds developed a wound infection post-BTM application. Two of these patients were treated for pre-existing infection, prior to BTM application—one progressed to major amputation and the other retained a chronic wound. The remaining patient progressed to complete wound healing within 182 days post-BTM application.

### 3.4. Pre-BTM Application Toe Pressures and Wound Healing Outcomes

Of the 13 patients with pre-BTM application toe pressures >50 mmHg, complete wound healing was observed in 11 (84%) patients, compared with the pre-BTM cohort with toe pressures of <50 mmHg (despite maximal revascularisation), where a lower proportion of patients (4 out of 10) completely healed their wounds.

There was a total of seven patients treated with BTM with severe refractory ischaemia post-maximal revascularisation (toe pressures < 30 mmHg). Of these seven patients, three (43%) progressed to complete wound healing, two (29%) retained chronic wounds and one (14%) progressed to minor amputation and one (14%) to major amputation within the observed period.

### 3.5. WIfI Stratification of 12-Month Amputation Risk

All 23 wounded limbs in 22 patients were successfully staged using the WIfI 12-month amputation risk. In total, 18 (78%) wounded limbs were scored a stage 4 (high) risk of amputation, 4 (17%) were assigned stage 3 (moderate) and 1 (5%) assigned a stage 1 (very low) score (Table 2). There were no patients assigned WIfI stage 2.

### 3.6. Post-BTM Application Wound Healing Outcomes and WIfI Stratification

Analysis of the wound healing outcomes in the 18 wounds assigned WIfI stage 4 observed complete wound healing in 61.1% (*n* = 11), with 11.1% (*n* = 2) retaining a chronic wound, 16.7% (*n* = 3) requiring minor amputation and 11.1% (*n* = 2) progressing to major amputation (Table 2).

There were four (17%) wounds stratified to WIfI stage 3. Complete wound healing was observed in 75% (*n*= 3) of this group, with the remaining wounded foot (25%) retaining a chronic wound. There were no amputations in this group (Table 2).

The single WIfI stage 1 wounded foot progressed to complete wound healing (Table 2). 

Stratification of time to complete wound healing to WIfI stage revealed averages of 166 days (SD 92.6) for WIfI stage 4 and 141 days (SD 25.5) for WIfI stage 3 patients. The single WIfI stage 1 patient healed their wound in 142 days (Table 1). 

### 3.7. Surgical Amputation and Chronic Wound Debridement Outcomes Stratified by WIfI Stage

The majority (69.5%) of wounds treated with BTM were existing chronic wounds (*n* = 16), with the remaining portion of wounds originating from surgical amputations (*n* = 7) (Table 3). All WIfI stages were represented in the chronic wound cohort, with the majority stage 4 (*n* = 11) followed by stage 3 (*n* = 4) and stage 1 (*n* = 1), respectively. Major amputation was undertaken in one (6%) and minor amputation in three (19%) of the sixteen patients in the chronic wound cohort, and all were classified WIfI stage 4 (Table 3). Wound healing was observed in 100% of WIfI stage 1 cases, and a majority of both WIfI stage 3 (75%) and stage 4 (63.3%) BTM-treated chronic wounds. Of the chronic wound patients treated with BTM, only one patient, classified as WIfI stage 3, retained an unhealed chronic wound at the 12-month observation period (Table 3).

All patients treated with BTM following surgical minor amputation were classified as WIfI stage 4, four (57%) were observed to progress to complete wound healing, two (29%) retained chronic wounds and only one (14%) required major amputation (Table 3).

### 3.8. Pre-BTM Application Glycosylated Haemoglobin

The average pre-procedural HbA1C of the 22-patient cohort was 9.0% (Median 8.5%, IQR 2.45%). Mean HbA1c values for patients with healed wounds (*n* = 15) were 9.5% (SD 2.5) compared with non-healed wounds (*n* = 8), which was 8.1% (SD 1.4). No significant difference in HbA1C was observed between the healed and non-healed groups (*p* = 0.17) (Table 1).

### 3.9. Safety

There were no adverse events related to BTM application reported during the observation period.

## 4. Discussion

Ulceration in the setting of diabetes-related foot disease (neuropathy, structural deformity and peripheral arterial disease) has been established as a major contributing factor to lower limb amputation [4].

A wide variety of studies exist detailing the use of various augmentative wound healing technologies for the treatment of diabetic foot ulceration, including dermal matrices (exogenous and human-derived), allografts, xenografts and autologous grafts [5]. Reported outcomes from these studies vary, with ulcer healing rates spanning from 30% to as high as 85% in their experimentally treated groups; however, many of these studies exclude patients with severe lower limb peripheral arterial disease, tissue or bone infection or exposed underlying structures, such as tendon or bone [17,18,19,20,21,22,23,24]. With a majority of WIfI stage 4 patients in our observational cohort, our use of BTM was employed to treat moderate to severe diabetic foot ulcerations and should be taken into account when evaluating the reported effectiveness of this technology. Despite the high severity, we observed overall healing rates of 65.3% and 12-month major amputation rates of 11.1% in our WIfI stage 4 cohort, comparable with those previously reported by Mathioudakis, Hicks [25], who, in their single-centre prospective observational study of diabetic foot wounds treated in a multidisciplinary setting, observed a 70% wound healing rate and 6% 12-month major amputation rate in their WIfI Stage 4 cohort. This comparison should be made in the context that under half (44%) of those WIfI stage 4 patients reported by Mathioudakis, Hicks [25], required surgical wound coverage (xenograft/allograft application), which is arguably a more accurately comparable cohort—however, outcome data for this subgroup were not reported. Further comparison of BTM efficacy can be drawn with our observed mean time to complete wound healing for WIfI stage 4 (166 days, SD 92) and stage 3 (141 days, SD 25), which were favourable when compared with previously published, single-centre prospective observational study data by Mathioudakis, Hicks [25] (stage 4, 190 days, and stage 3, 125 days), and Zhan, Branco [26] (stage 4, 264 days, and stage 3, 163 days).

When we consider that reported systematic review data suggest 12-month major-amputation risk of WIfI stage 3 and 4 patients to be 11% and 38% and we have shown amputation rates to be 0% and 11.1%, respectively, this highlights the potential future role that BTM has to play in the paradigm of such patients who may otherwise face amputation [16].

The higher rate of wound healing observed in the patient cohort with toe pressures >50 mmHg aligns with previously reported experiences [15]; however, this should not detract from the high severity of wounds treated with BTM, represented in this subgroup (Table 1), with 92% of those with a toe pressure >50 mmHg containing wounds scored as a WIfI wound category two or greater (extensive/deep ulcer, with exposed bone joint or tendon) (Table 1). Whilst severe refractory ischaemia (toe pressures < 30 mmHg post-maximal revascularisation) was prevalent in only 30% (*n* = 7) of the study cohort, with a majority (*n* = 6) of these cases hosting severe wounds (WIfI wound severity of 3), 12-month amputation-free survival was 70% (*n* = 5), suggesting BTM offers some capacity to facilitate amputation prevention in the setting of ischaemia not amendable to further revascularisation. These data highlight that the WIfI classification of our observed cohort was predominantly driven by high wound severity and, therefore, retain a high risk of wound healing failure, irrespective of perfusion status, reinforcing the positive context of our patient outcomes for which BTM offered a non-autologous off-the-shelf reconstruction solution.

Adding further support to the value of BTM is the fact that it is dry storage, non-refrigerated, low cost and does not result in donor site creation, which overcomes many of the current criticisms of skin grafts and tissue replacements, including high cost in the context of operating theatre time, creation of a second wound in a patient with established wound healing difficulties and variance in availability (in context of allografts/exografts) [5].

The reported rates of surgical site infection post-BTM application, while seemingly high, are in keeping with reported experiences of higher-than-normal infection rates in diabetic foot and ankle surgery (reported as 7.7% to 13.6%) [27]. Our observed cohort also had a considerably high number of pre-existing infections diagnosed prior to BTM application (Table 1). Despite this, our amputation rates were considerably lower than those reported in the published systematic review data [16].

It is also acknowledged that all patients who progressed to amputation, post-BTM application, were amongst those treated for infection pre-application of BTM. Although the sample size does not allow for multivariate analysis in this cohort, infection source control through aggressive debridement and antibiotics is, therefore, recommended prior to BTM application.

While WIfI provides a vehicle for describing neuroischaemic disease burden, glycaemic control is also a key influence in wound healing. Pre-procedure HbA1C failed to establish a significant link between those wounds that healed and those that failed to heal within the follow-up time frames. This crudely suggests an equal distribution of diabetes disease severity amongst the observed population—however, this conclusion could have been further strengthened through the gathering of additional data, such as type of blood-glucose-regulating medication, as additional indicators of diabetic disease severity. While our findings do not align with emerging evidence associating elevated HbA1C with increased amputation risk and decreased rates of wound healing, this is more likely a reflection of the low case volume and/or other comorbid factors not yet identified [28,29].

This prospective study presents encouraging results, with respect to limb salvage and wound healing, for patients with moderate to severe diabetic foot ulceration treated with BTM. This extent of diabetic foot ulceration being successfully treated with dermal substitutes is scarcely reported in the literature and reinforces a major role for BTM in this cohort. Furthermore, our observational study presents a real-world pragmatic experience of the use of BTM in treating very advanced diabetic foot disease and overcomes the criticisms of many such papers who do not report on such advanced disease [16,17,22,23,24].

It is noted that one limitation of BTM application relates to the need for apposition of the foam with all surfaces of the wound bed and, therefore, some wound geographies, such as deep narrow cavities, exclude the use of BTM in its current format.

While early experience with BTM in human studies suggests that the product facilitates neovascularisation into the foam scaffold to allow for integration and proliferation of a neodermis, including coverage of exposed deeper tissue structures [7,8], its specific action in diabetic foot disease is not established and requires further investigation. Our early experience reinforces the ability for BTM to provide coverage of deep tissue structures in complex diabetic foot wounds in whom the alternative option may only be major limb amputation.

Direction for future adaptation of this technology would be best informed through linking wound healing outcomes with tissue-level laboratory-based investigation of the impact of BTM on the cellular milieu of diabetic foot ulcers and our institution is currently undertaking this work.

## 5. Conclusions

The results from this observational study serve as encouraging early data for the use of BTM as a safe and accessible treatment of neuroischaemic diabetic foot wounds, including those that are moderate or severe. While larger-scale data are required, BTM may represent a promising new addition to the armamentarium of clinicians, who strive to achieve limb salvage in this complex cohort of patients.

## Figures and Tables

**Figure 1 biomedicines-11-00721-f001:**
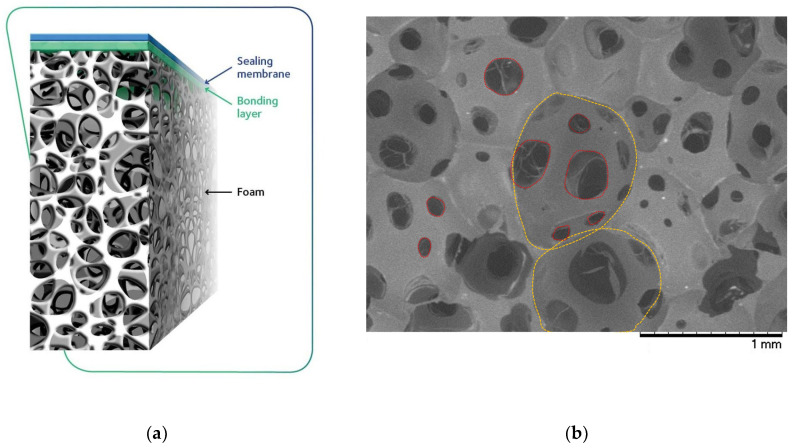
(**a**) Schematic of NovoSorb BTM showing the biodegradable polyurethane foam matrix bonded to the non-biodegradable polyurethane sealing membrane; (**b**) scanning electron microscope image showing microarchitecture of NovoSorb BTM. Chambers (yellow) are connected by pores (red) that vary in size, with an average of ~188 µm. (© PolyNovo Biomaterials Pty Ltd., Used with permission.).

**Figure 2 biomedicines-11-00721-f002:**
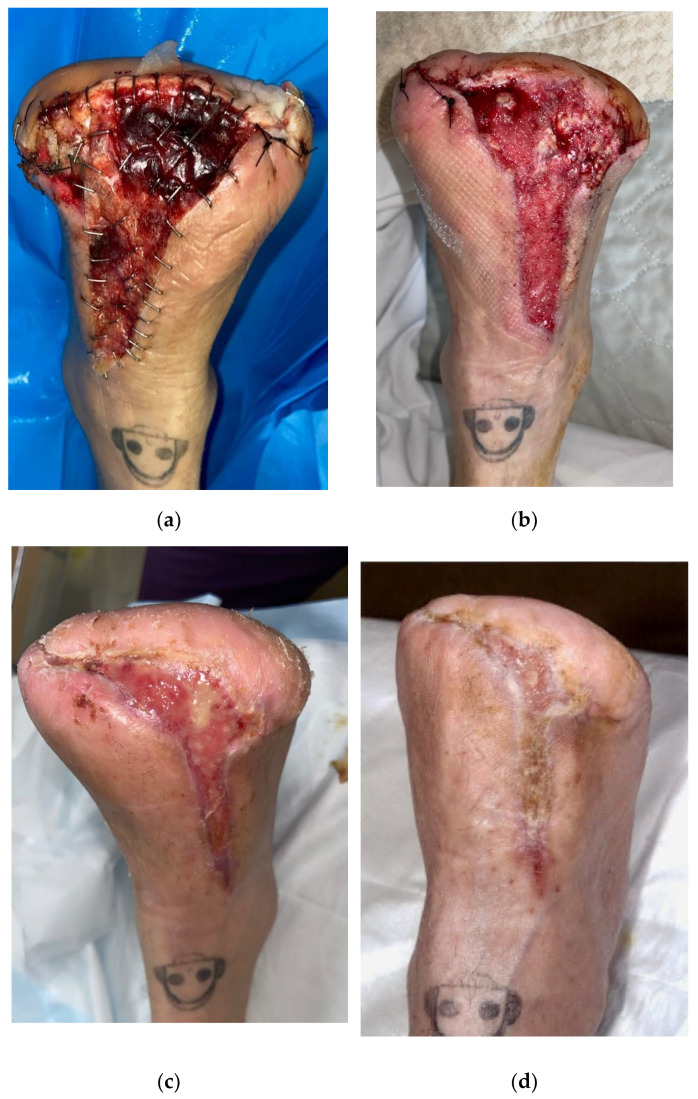
(**a**) Left forefoot amputation wound, day 28 post-transmetatarsal amputation, day 10 post-BTM application; (**b**) day 24 post-BTM application, sealing membrane removed to reveal healthy granulation tissue; (**c**) day 45 post-BTM application, contraction of wound bed and advancing epithelial margins; (**d**) day 55 post-BTM application, significant surface area reduction, only small area of open wound bed remains.

**Table 1 biomedicines-11-00721-t001:** Demographics and case characteristics.

Descriptors	(*n*%)
Participants	22 (100%)
Wounds treated with BTM *	23 (100%)
Total BTM applications *	25
Age (years)	67.3 (SD 10.1)
Sex Male (%)	19 (86.3%)
Median HbA1c% (mmol/mol) †	8.5% (69) IQR 2.45%
WIfI Categories ‡	
Wound	
Category 1	1
Category 2	4
Category 3	18
Ischaemia (Toe Pressures)	
≥60 mmHg	11
40–59 mmHg	5
30–39 mmHg	0
<30 mmHg	7
Foot Infection (Pre-BTM *)	
Uninfected	11
Mild Infection	4
Moderate	1
Severe	7
Average Time to complete wound healing (days)—full cohort	158 (SD 94)
Average time to complete wound healing—WIfI ‡ stage 4	166 (SD 92)
Average time to complete wound healing—WIfI ‡ stage 3	141 (SD 25)

* Bilayer biodegradable synthetic matrix tissue scaffold technology (BTM). † Glycosylated haemoglobin (NGSP HbA1C). ‡ Wound, Ischaemia and foot Infection classification system (WIfI).

**Table 2 biomedicines-11-00721-t002:** WIfI * stratification versus outcomes at 12 months post-BTM † application.

WIfI * Stratification (1-Year Amputation Risk)				
Wound Outcome	Stage 4 (High Risk)	Stage 3 (Moderate Risk)	Stage 1 (Very Low Risk)	Totals
Wound Healed	11 (61.1%)	3 (75%)	1 (100%)	15 (65.3%)
Chronic Wound	2 (11.1%)	1 (25%)	0	3 (13%)
Minor Amputation (Post BTM †)	3 (16.7%)	0	0	3 (13%)
Major Amputation	2 (11.1%)	0	0	2 (8.7%)
Totals	18 (78%)	4 (17%)	1 (5%)	23 (100%)
% Amputation free survival @ 12 months	83% (15/18)	100% (0/4)	100% (0/1)	

* Wound, Ischaemia and foot Infection classification system (WIfI). † Bilayer biodegradable synthetic matrix tissue scaffold technology (BTM).

**Table 3 biomedicines-11-00721-t003:** Wound healing outcomes stratified by wound origin and WIfI * stage.

Wound Origin	WIfI * Stage	Outcome			
		Healed	Chronic	Amputation	Totals
Chronic Wound	Very Low (1)	1	0	0	1
	Moderate (3)	3	1	0	4
	High (4)	7	0	4	11
Surgical (Post-Amputation)	High (4)	4	2	1	7
Totals		15	3	5	23

***** Wound, Ischaemia and foot Infection classification system (WIfI).

## Data Availability

The data presented in this study are available on request from the corresponding author. The data are not publicly available due to the privacy of research subjects.

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
