# Peer review of "Overcoming Barriers to Wound Healing in a Neuropathic and Neuro-Ischaemic Diabetic Foot Cohort Using a Novel Bilayer Biodegradable Synthetic Matrix"

_biomedicines, 2023, doi:10.3390/biomedicines11030721_

Round 1

Reviewer 1 Report

This manuscript was well organized and written. Some minnor revisions are suggested as follows:

(1) Some important characterization inforamtion about the bilayer synthetic matrix are missing. For instance, the actual photograph, SEM images of the dressing materials are suggested to be prestented, in order to increase the readability. In addition, how about the cross-sectional structure and mechanical properties of the dressings?

(2) The current limitation and future prospectives using these novel dressing materals are suggested to be added. 

Author Response

Thank you for your review of our manuscript and insights and suggestions for improvement. Please see the attachment for our responses.

Reviewer 2 Report

The authors review their experience using BTM to treat diabetic foot wounds. Overall, the report is interesting and well-done. I have a few minor comments:

1) BTM typically is grafted when it is vascularized. Why did you not graft the wounds to close them faster?

2) It seems that healing at 1 year is fairly late. Why not look at healing at an earlier time point?

3) Are you considering a prospective trial comparing BTM to no BTM?

4) On line 105, you spell "was" as "wass". 

Author Response

Thank you for reviewing our manuscript and for your suggestions for improvement. Please see the attachment for our responses and outline of changes. 
